# Were changes in stress state responsible for the 2019 Ridgecrest, California, earthquakes?

K. Z. Nanjo [1,2,3✉]

Monitoring the Earth's stress state plays a role in our understanding of an earthquake's mechanism and in the distribution of hazards. Crustal deformation due to the July 2019 earthquake sequence in Ridgecrest (California) that culminated in a preceding quake of magnitude ($M$) 6.4 and a subsequent $M$7.1 quake caused stress perturbation in a nearby region, but implications of future seismicity are still uncertain. Here, the occurrence of small earthquakes is compared to larger ones, using $b$-values, showing that the rupture initiation from an area of low $b$-values, indicative of high stress, was common to both $M$6.4 and $M$7.1 quakes. The post-$M$7.1-quake sequence reveals that another low-$b$-value zone, which avoided its ruptured area, fell into an area near the Garlock fault that hosted past large earthquakes. If this area were more stressed, there would be a high-likelihood of further activation of seismicity that might influence the Garlock fault.

[1] Global Center for Asian and Regional Research, University of Shizuoka, 3-6-1, Takajo, Aoi-ku, Shizuoka 420-0839, Japan. [2] Center for Integrated Research and Education of Natural Hazards, Shizuoka University, 836, Ohya, Suruga-ku, Shizuoka 422-8529, Japan. [3] Institute of Statistical Mathematics, 10-3 Midori-cho, Tachikawa, Tokyo 190-8562, Japan. ✉email: nanjo@u-shizuoka-ken.ac.jp

The 2019 Ridgecrest earthquakes, which occurred near the town of Ridgecrest, California, included a magnitude ($M$) 7.1 quake that struck on 5 July 2019 (UTC) as well as active foreshocks and aftershocks[1] (Fig. 1a). A $M$6.4 event preceded the $M$7.1 quake 34 h later. The $M$7.1 quake ruptured the Earth's surface and involved a right-lateral strike slip along a NW-SE trending fault. A predominant mechanism of the $M$6.4 quake was a left-lateral strike-slip fault motion along a NE-SW trending fault that conjugated with the fault of the $M$7.1 quake. The broad context of the Ridgecrest earthquakes is that they occurred under the current tectonic stress that created the Eastern California Shear Zone (ECSZ), a seismically active region east of the southern segment of the San Andreas Fault[2]. The $M$6.4 quake was followed by more than 1000 perceivable events until the $M$7.1 quake. A post-$M$7.1-quake sequence is still active. Over the first 8 months since the $M$7.1 quake, about 30,000 events with $M \geq 1$ occurred, including more than 90 events with $M \geq 4$.

Crustal deformation due to the occurrence of large earthquakes causes stress perturbation in nearby regions. From the viewpoint of the physics of earthquakes, the probability of a subsequent large earthquake depends on the stress conditions set up by the previous events and long-term tectonic state[3]. Given the tectonic stress of the ECSZ, an investigation into the spatio-temporal state of stress along and near the faults coseismically ruptured by the $M$7.1 and $M$6.4 quakes can play a crucial role in understanding the distribution of post-seismic hazards after these quakes. Coulomb stress models were used to explain that the site of the $M$6.4 quake was stressed by the great 1872 Owens Valley ($M\sim$7.6), the 1992 Landers ($M$7.3), and the 1999 Hector Mine ($M$7.1) quakes, and that the $M$6.4 earthquake loaded the site where the $M$7.1 shock nucleated[4–6]. However, physics-based approaches employing Coulomb stress transfer have so far not been successful in forecasting upcoming large earthquakes any better than statistical models[7]. This is partly due to the fact that the locations of potential faults, essential inputs to the calculation of change in Coulomb stress, are unknown[8].

An alternative statistics-based approach is used to infer changes in the stress state, focusing on the fact that the Ridgecrest earthquakes occurred within the seismically active ECSZ, with data of good enough quality and sufficient abundance, collected by the SCSN (Southern California Seismic Network)[9] (for details on the earthquake dataset, see Methods). This approach uses a statistical model based on seismicity: the $b$-value of the Gutenberg–Richter (GR) law[10], given as $\log_{10}N = a - bM$, where $N$ is the cumulative number of earthquakes with a magnitude larger than or equal to $M$, $a$ characterizes seismic activity or earthquake productivity of a region, and the constant $b$ is used to describe the relative occurrence of large and small events (i.e., a high $b$-value indicates a larger proportion of small earthquakes, and vice versa). The $b$-value is sensitive to differential stress, and its inverse dependence on differential stress has been confirmed many times in both laboratory and field studies[11–15] (for details on the $b$-value estimation, see "Methods").

Here, earthquake triggering and characteristics of seismicity before, during, and after the Ridgecrest earthquakes are investigated. In particular, focus is placed on determining maps of $b$-values for different time periods, showing how the nucleation area for both the $M$6.4 and $M$7.1 quakes had low $b$-values before these events occurred, and mid-to-high $b$-values thereafter. The $b$-value map also correlates well with the slip distribution of the $M$7.1 quake. In addition, the local and time-dependent variations in $b$-values of the Ridgecrest earthquakes are linked with estimates of changes to Coulomb stress. The main conclusions of this study are that the $b$-values provide insight into the state of stress in the fault zone, which is likely closely related to the nucleation and evolution of earthquakes in the sequence. This combined approach of $b$-value and stress-change analyses to the post-$M$7.1-quake seismicity shows an area that is currently being stressed. Monitoring the spatio-temporal distribution of $b$, together with other seismological and geodetic observations, will contribute to an appreciation of the seismic hazard in the ECSZ.

## Results

**Temporal variation associated with stress changes**. Different periods were considered to find time-dependent signals that are consistent with stress increase and release. Two periods that separate data before and in between the two large quakes were selected: first, before the $M$6.4 quake; and second, 34 h after the $M$6.4 quake up to the $M$7.1 quake. For inference on the distribution in seismic hazards, another period that is about eight months after the rupture of the $M$7.1 quake until 23 March 2020 will be discussed later.

**Pre-$M$6.4-quake sequence**. A map view (Fig. 1b) based on seismicity before the $M$6.4 quake with a depth range of 7–13 km shows a zone of low $b$-values ($b \sim 0.6$) around the future hypocenter of depth 10.7 km (for details on the mapping procedure, see Methods and Supplementary Figs. 1 and 2). Shallow seismicity (depth of 0–7 km) shows no clear zone of such low $b$-values near the future epicenter (inset of Fig. 1b). The low-$b$-value zone was seen, even when the $M$4.0 quake and its following events that occurred during the last 30 minutes before the $M$6.4 quake near the eventual hypocenter were excluded from the mapping (Supplementary Fig. 2d). For earthquakes around the $M$6.4 epicenter (Fig. 1c), the $b$-values were mostly above 1 until 2010. Since 2010, the $b$-values have shown a gradual decrease over time, to values near 0.7. The final values are remarkably similar to those immediately before the entire fracture, as was obtained in a previous laboratory experiment[11].

The $M$6.4 quake ruptured conjugate faults: the 6-km-long northwest-trending fault first slipped, followed by a slip in the ~15-km-long southwest-trending fault[1,2]. The initial portion of the $M$6.4 quake terminated about 4 km from the eventual $M$7.1 hypocenter. This 4-km gap was progressively filled by a series of moderate-sized earthquakes in the 34 hours after the $M$6.4 quake, which suggests that this portion of the fault acted as a barrier through which the $M$6.4 rupture was unable to propagate[1]. This was confirmed by the cross-sectional views (Fig. 2a, b) for the pre-$M$6.4-quake period (see Methods and Supplementary Figs. 3–5 for the mapping procedure). Low $b$-values ($b < 0.9$: purple to blue) were seen near the $M$6.4 hypocenter, while high $b$-values ($b > 1$: yellow to orange) were seen near the $M$7.1 hypocenter. This was interpreted as an indication of a weakly stressed area into which the $M$6.4 rupture was not allowed to propagate.

**Pre-$M$7.1-quake sequence**. The distribution of $b$-values (Fig. 2c, d) based on seismicity during a period before the $M$7.1 quake, indicated by the bidirectional arrow in the inset of Fig. 2c, shows a zone of low $b$-values near the eventual $M$7.1 hypocenter. A comparison with the pre-$M$6.4-quake period in Fig. 2a, b shows that an increase in $b$ at the $M$6.4 hypocenter and a decrease in $b$ at the $M$7.1 hypocenter are significant (Fig. 2g). The result indicates that the $M$6.4 rupture relaxed stress near the $M$6.4 hypocenter, which had been highly stressed before the $M$6.4 quake, but that it transferred stress to the nearby region of the $M$7.1 hypocenter, which had acted as a barrier before the $M$6.4 quake. The result was the erosion of this barrier by seismicity.

To confirm that this erosion triggered the $M$7.1 quake, Coulomb stress transfer was calculated[16,17] (for details on the fault models and the stress-change calculation, see Methods), revealing that a region around the hypocenter of the $M$7.1 quake

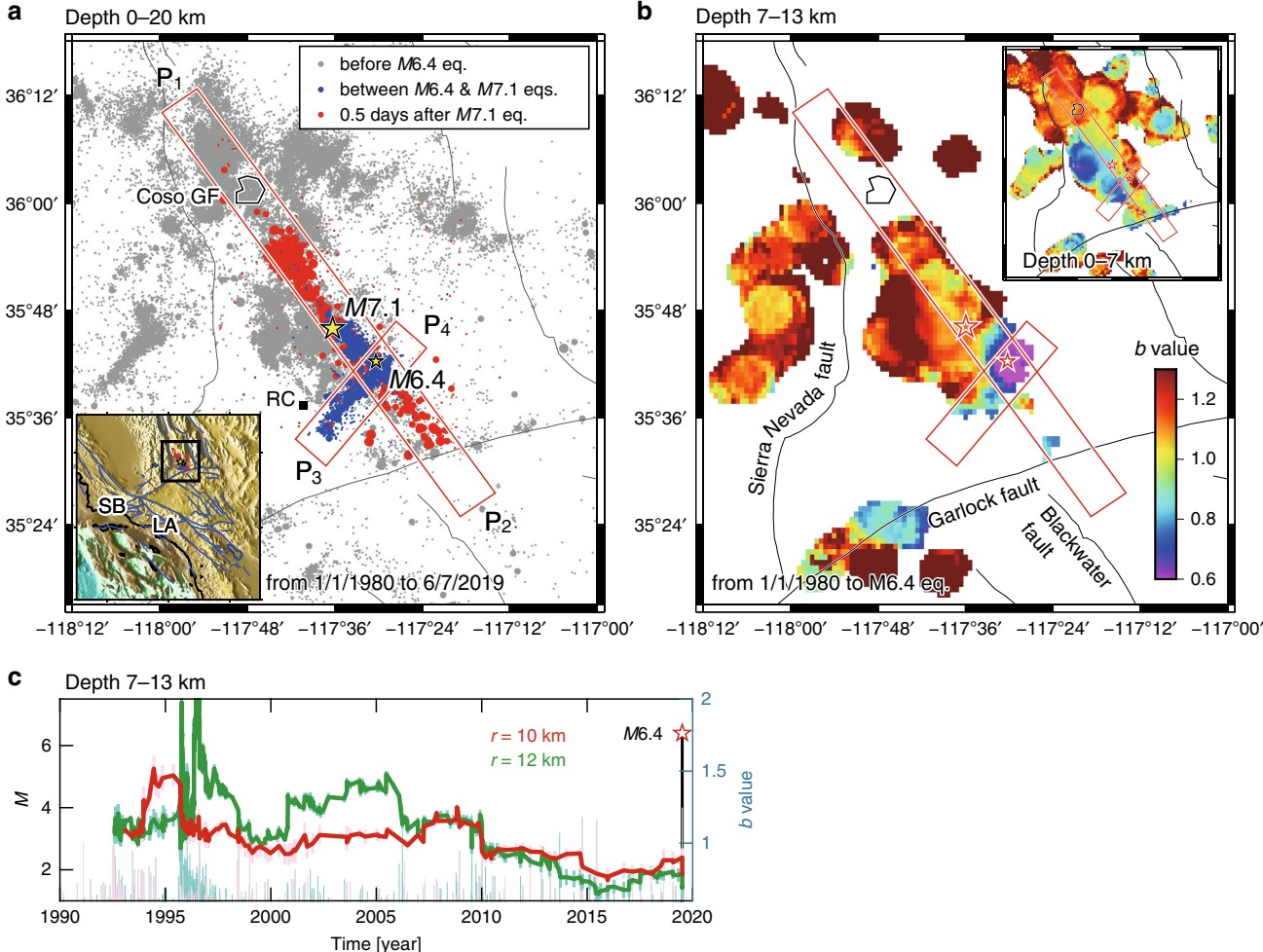

**Fig. 1 Ridgecrest earthquake sequence and *b*-values. a** Map of earthquakes in the Ridgecrest region. The cross-section in Fig. 2 extends from P₁ to P₂ and from P₃ to P₄ with a width of 8 km. Inset shows the study region (black rectangle). Thin lines indicate major mapped faults. Los Angeles, Santa Barbara, Ridgecrest, and Coso geothermal field are indicated as LA, SB, RC, and Coso GF, respectively. **b** Map of *b*-values obtained from seismicity ($M \geq 1$) at a depth of 7-13 km before the $M6.4$ quake. Inset: the *b*-value map at a depth of 0-7 km. **c** Plot of *b* as a function of time before the $M6.4$ quake for seismicity (depth of 7-13 km) falling in the circle with a radius of $r = 10$ km (red) and 12 km (green), centered at the $M6.4$ epicenter. Moving windows cover 100 events. Also included is the magnitude-time dependence.

became about 2 bars closer to failure by the $M6.4$ quake and its subsequent seismicity (Fig. 3a). To show this map, faults of the $M6.4$ quake and the relatively large events until immediately before the $M7.1$ quake were assumed as source faults (Supplementary Fig. 6). For the $M6.4$ quake, only the southwest-trending fault was assumed. This is because a large slip of the $M6.4$ quake occurred along the southwest-trending fault rather than along the conjugate northwest-trending fault. The former fault (~15 km long) is much longer than the latter (6 km long). A comparison with a case that only considered the $M6.4$ quake as a source fault (inset of Fig. 3a) shows that the large changes in Coulomb stress near regions of the $M7.1$ hypocenter were very likely due to the $M6.4$ quake as well as its subsequent earthquakes[18]. Even if the conjugate faults of the $M6.4$ quake were assumed as source faults, stress in the region near the $M7.1$ hypocenter increased[5].

Additional insight into changes in the stress state was provided by temporal behavior of the sequence following the $M6.4$ quake. Relatively large events occurred early in the post-$M6.4$-quake sequence (grey stem plot in the inset of Fig. 2c), and the mean magnitude of these events evolved into small values over time. This behavior is well modeled by the Omori-Utsu (OU) power-law aftershock decay[19], given as $\lambda \sim t^{-p}$, where $t$ is the time since the occurrence of a mainshock; $\lambda$ is the number of aftershocks per

unit time at $t$ with a magnitude greater than or equal to a cutoff magnitude; and $p$ is a constant (for details on the OU law, see Methods). $p = 1$ is a good approximation[20], but spatio-temporal changes in $p$ are observable. $M \geq 3$ events were used, taking homogeneity of seismicity recordings into consideration (see Methods and Supplementary Fig. 7 for homogeneity of seismicity recordings). Modeling these events showed that $p$ was smaller for the northern area, including the $M7.1$ hypocenter, than for the southern area (Fig. 4), revealing that decay in seismicity was slower in the former area than in the latter one (see also Supplementary Fig. 8). This result is interpreted as an indication of a slower decrease in stress in the northern area than in the southern area, according to fictional theory[21]. This supports the result of a *b*-value map before the $M7.1$ quake (Fig. 2c, d) that showed lower *b*-values (indicative of higher stress) in areas near the $M7.1$ hypocenter than in areas near the $M6.4$ hypocenter.

Low *b*-values near the $M7.1$ hypocenter (Fig. 2c, d), together with a temporal decay in seismicity (Fig. 4 and Supplementary Fig. 8), closely match another observation of increased Coulomb stresses near the $M7.1$ hypocenter (Fig. 3a). The sequence of stress jumps caused by the $M6.4$ quake and its subsequent events resulted in an increase of roughly 2 bars. This value is not

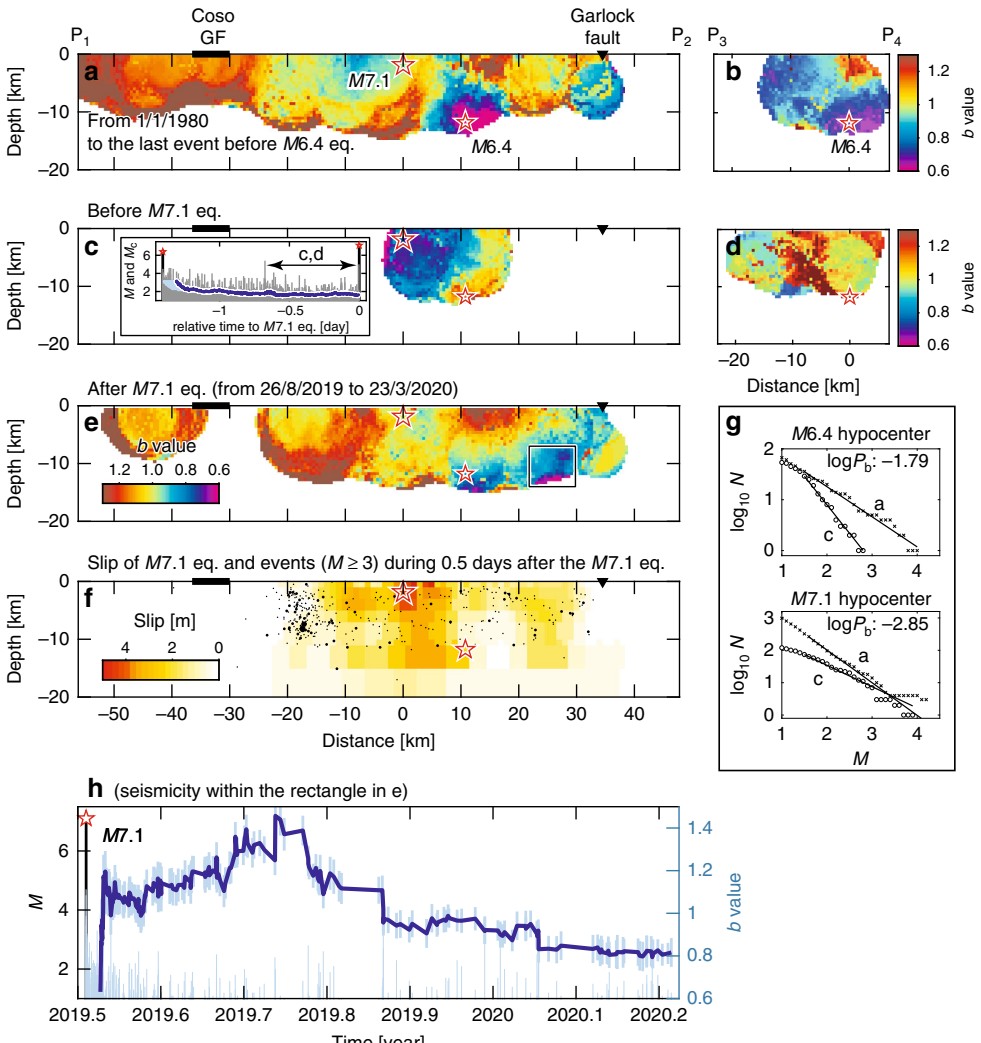

**Fig. 2 Cross-sectional views of *b*-values. a** *b*-values for seismicity ($M \geq 1$) before the *M*6.4 quake along the fault ruptured by the *M*7.1 quake. Stars shows the *M*7.1 and *M*6.4 hypocenters. **b** Same as **a** for the cross-section along the fault ruptured by the *M*6.4 quake. **c, d** Same as **a** and **b** for seismicity before the *M*7.1 quake. *b*-values were calculated for the period indicated by c, d in the inset of **c** from the first event after the *M*5.4 quake at a relative time of −0.672 days to the last event before the *M*7.1 quake. The use of seismicity soon after the *M*6.4 quake was avoided to remove the effect of strong temporal variability in *b*. Inset: plot of *M* and completeness magnitude ($M_c$) as a function of time relative to the *M*7.1 quake (see Supplementary Fig. 7). **e** Same as (**a**) for seismicity after the *M*7.1 quake. Events during the period from immediately after the *M*7.1 quake to 25 August 2019 (or 2019.65 decimal years) were not used to calculate *b*-values for the same reason as (**c, d**). **f** Slip distribution of the *M*7.1 quake[23], and events ($M \geq 3$) that occurred in the first 12 h. Symbol size is proportional to magnitude. **g** Top panel: frequency-magnitude distribution of earthquakes falling within a cylindrical volume with a 5-km radius, centered at the location of the *M*6.4 hypocenter in (**a** and **c**): a with $a = 2.44$, $b = 0.59 \pm 0.17$, and $M_c = 1.5$ and c with $a = 3.14$, $b = 1.12 \pm 0.33$, and $M_c = 1.5$ (for details on *b*-value estimation, see Methods). Bottom panel: same as the top one for the location of the *M*7.1 hypocenter: a with $a = 4.03$, $b = 1.01 \pm 0.07$, and $M_c = 1.6$, and c with $a = 2.86$, $b = 0.66 \pm 0.22$, and $M_c = 1.5$. Values of $\log P_b \leq$ -1.3 indicate a significant difference in $b$[48]. **h** Plot of *b* as a function of time after the *M*7.1 quake for seismicity within the rectangle in (**e**). Plotting procedure is the same as that for Fig. 1c.

surprising and is comparable to that obtained in previous studies[2,5].

To support the observation that the events preceding the *M*7.1 quake very likely played a role in triggering the eventual *M*7.1 event, an independent analysis from the above stress-related analyses was conducted. This was achieved by investigating if any sign indicative of the *M*7.1 quake could be found in the spatial organization in seismicity after the *M*6.4 quake (for details on spatial organization, see Methods). According to a previous study[22], the spatial concentration of smaller magnitude events (retrospectively named foreshocks) near the eventual event (retrospectively named mainshock) was a common feature of large earthquakes in southern California. To examine whether this was observed for the *M*7.1 quake, the quantity $\phi = R^{-1}/R_b^{-1}$

was selected[22], where $R^{-1}$ represents the inverse distance from position *x* to an event that occurred before a given time, averaged over the last *n* events before this given time, and $R_b^{-1}$ is the same as $R^{-1}$ but the average is taken over the second-to-last *n* events. $\phi > 1$ indicates a concentration of seismicity before the given time in an area surrounding *x*, and $\phi < 1$ indicates the dispersion of seismicity. A cross-sectional view (Fig. 5) of $\phi$-values with $n = 25$ (a typical value for southern California[22]) at the time immediately before the *M*7.1 quake shows a region of seismic concentration ($\phi \sim 1.5$) near the hypocenter of this quake. Similar to the *p*-value analysis, $M \geq 3$ events were used for the $\phi$-value calculation. Near the future *M*7.1 hypocenter, there was a gradual increase in $\phi$ to values above 1, while in other regions, $\phi$-values showed low values or a decreasing trend to values of $\phi \sim 1$ or below 1. These

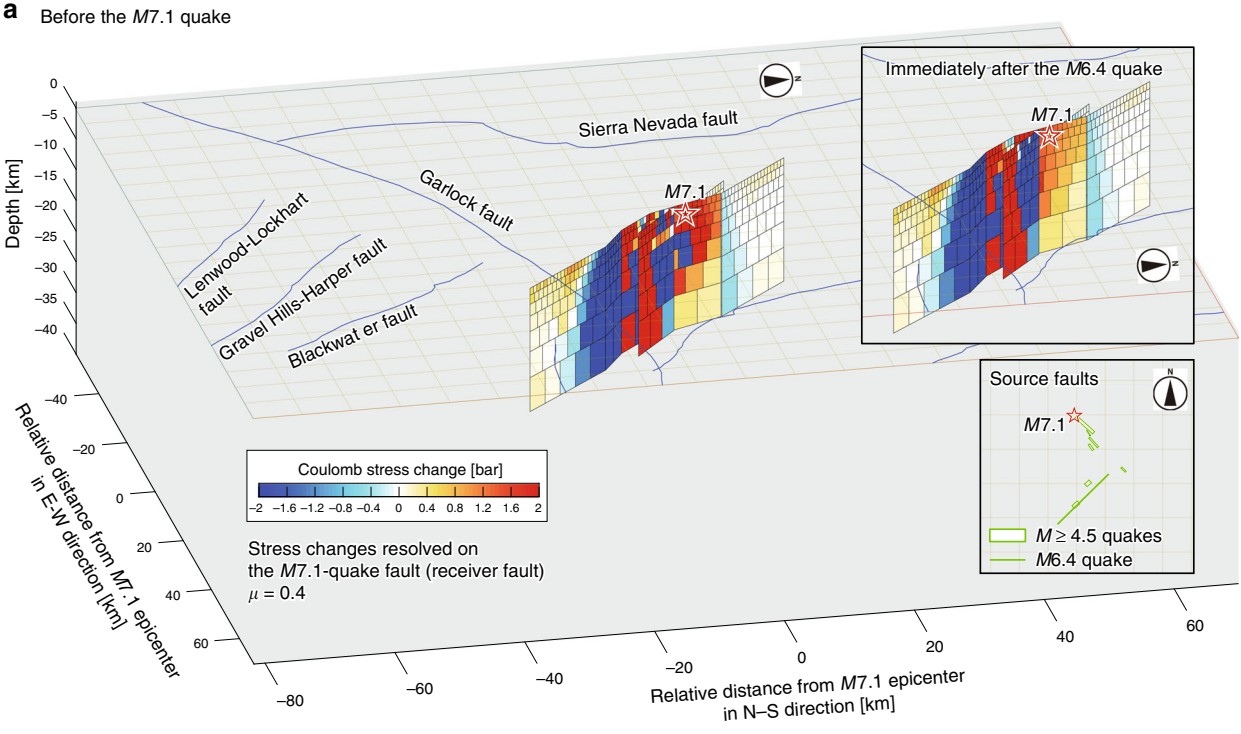

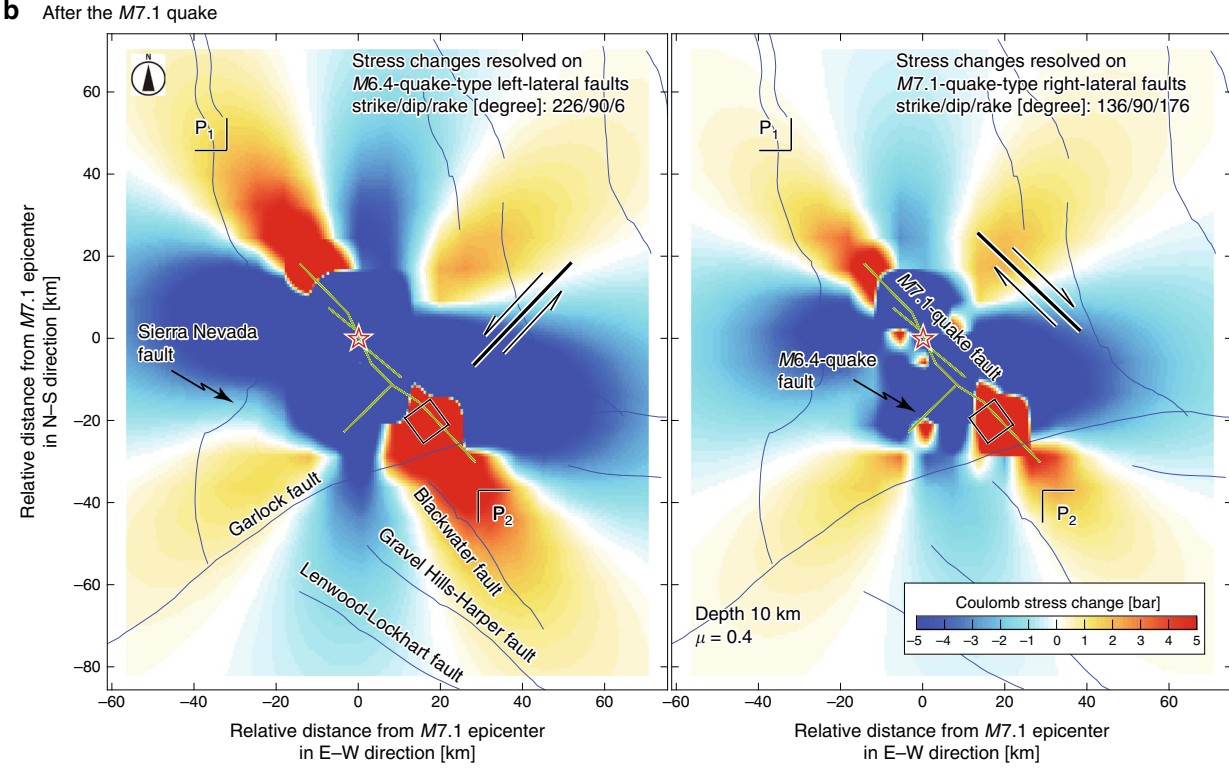

**Fig. 3 Coulomb stress changes. a** Stress changes resolved on the *M*7.1 quake fault as a result of the *M*6.4 quake and the following *M* ≥ 4.5 events. Star indicates the *M*7.1 hypocenter. Top inset: changes in Coulomb stress as a result of only the *M*6.4 earthquake, showing that the increase in stress near the region of the *M*7.1 hypocenter was as high as 1 bar (orange). Bottom inset: source faults projected on the Earth's surface. The *M*6.4-quake fault is indicated by a segment because it is assumed to be a vertical plane. Rectangles indicate fault planes of *M* ≥ 4.5 events. For details on fault models, see Methods. **b** Changes in stress at a depth of 10 km as a result of the *M*6.4 and *M*7.1 quakes. Green segments indicate source faults (*M*6.4 and *M*7.1 quakes). Left panel: changes in stress resolved on *M*6.4-quake-type left-lateral faults (black line with a half-arrow pair). Right panel: changes in stress resolved on *M*7.1-quake-type right-lateral faults. The rectangle indicates an area of low *b*-values shown by the rectangle in Fig. 2e that displays the cross-section extending from P₁ to P₂. See Supplementary Figs. 11 and 12 for other depths.

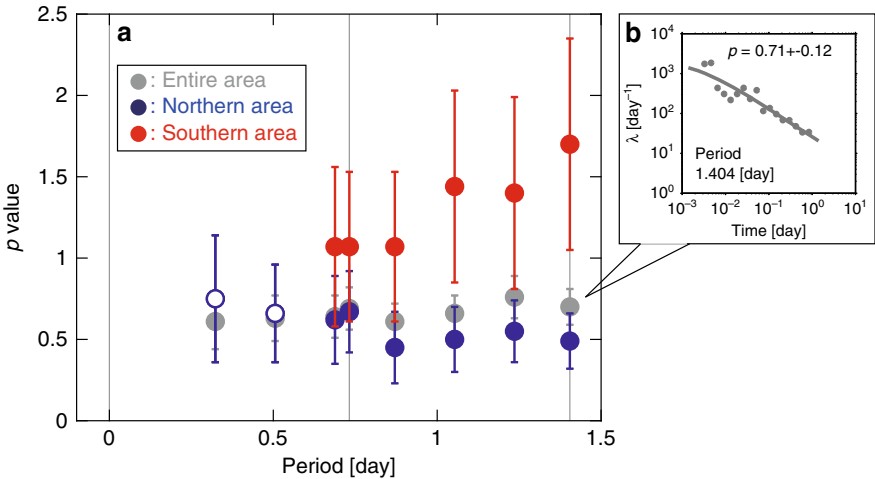

**Fig. 4 Fitting of the OU law. a** Plot of *p*-value of the OU law as a function of the length of the analyzed period since the *M*6.4 quake, based on seismicity ($M \geq 3$) during the period between *M*6.4 and *M*7.1 quakes along the fault such that the *M*7.1 ruptured in the entire area (grey), in the northern area (North of 35.72°N) (blue), and in the southern area (South of 35.72°N) (red). The maximum-likelihood fit was used to determine a *p*-value. Uncertainties in *p* were computed by bootstrapping. Open circles for the northern area show *p*-values obtained based on $N \leq 20$ earthquakes. For the periods $\leq 0.5$ days, no *p*-value was obtained for the southern area, because the solution did not converge due to not enough data analyzed. Vertical lines indicating the periods of 1.404, 0.732, and 0 days correspond to the periods ending at the time of the *M*7.1, *M*5.4, and *M*6.4, quakes, respectively. **b** Number λ (day$^{-1}$) of seismicity ($M \geq 3$) as a function of time from the *M*6.4 quake for the analyzed period of 1.404 days in the entire area (grey).

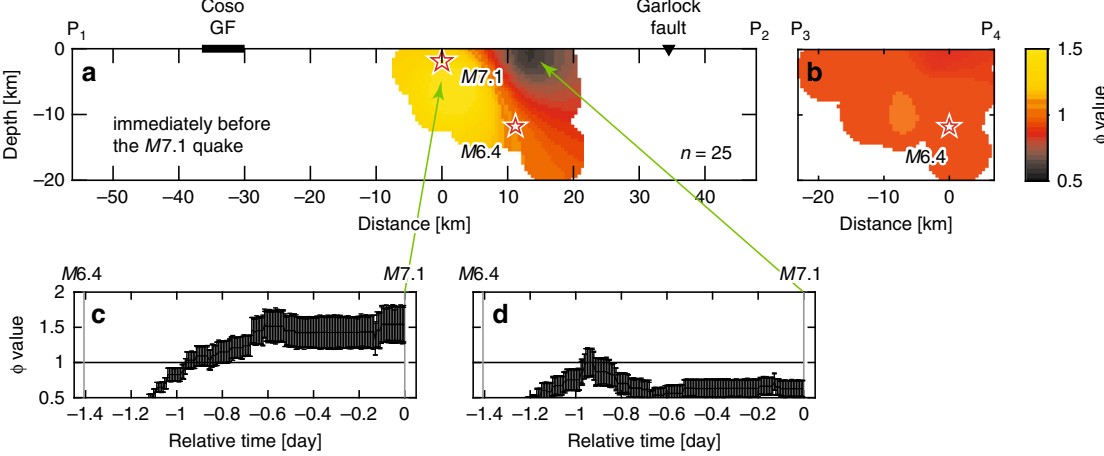

**Fig. 5 Seismicity concentration/dispersion. a** φ-values with $n = 25$ at the time immediately before the *M*7.1 quake, based on seismicity ($M \geq 3$) during the period between *M*6.4 and *M*7.1 quakes along the fault ruptured by the *M*7.1 quake. Note that seismicity used includes from the first event after the *M*6.4 quake to the last event before the *M*7.1 quake. Stars indicate the *M*6.4 and *M*7.1 hypocenters. **b** Same as **a** for the cross-section along the fault ruptured by the *M*6.4 quake. (**c**, **d**) Plot of φ as a function of relative time to the *M*7.1 quake at the locations indicated by arrows. Uncertainties in φ, used to draw error bars in **c** and **d**, were computed by bootstrapping (see also Supplementary Fig. 10). The occurrence time of the *M*6.4 quake, relative to that of the *M*7.1 quake, is indicated by a grey vertical line at −1.404 days. The relative time of the *M*7.1 quake is 0 days (grey vertical line), as is obvious. See Supplementary Fig. 9 for other *n*-values.

results depend weakly on *n* for $n = 15$-35 (Supplementary Figs. 9 and 10), as was observed in a previous study[22]. Our results show that the spatial organization of the pre-*M*7.1-quake sequence in a region near the eventual hypocenter was similar to that observed for previous southern California earthquakes[22], but it was dissimilar in other regions. This probably reflects the erosion by active seismicity toward a region near the *M*7.1 hypocenter. Thus, the spatial clustering before the *M*7.1 quake was a foreshock-type one indicative of a future mainshock, supporting the observation based on the above stress-related analyses.

**Post-*M*7.1-quake sequence.** The *M*7.1 quake nucleated about 10 km to the northwest of the *M*6.4 event and its rupture propagated

bilaterally, where most slips occurred near the *M*7.1 hypocenter[23]. The pre-*M*6.4-quake *b*-values (Fig. 2a) were compared with the slip distribution of the *M*7.1 quake[23] (Fig. 2f), showing peak-slip values of 4–5 m around the *M*7.1 hypocenter (relative distance of −2 to 5 km with depth of −5 to 0 km). It was found that this peak-slip area did not overlap with high *b*-values ($b > 1.1$: indicative of low stress), a feature that is common to many other earthquakes[24–26]. The influence of structural heterogeneity on the spatial distribution of *b*-values was also noted in such a way that rupture propagation of the *M*7.1 quake to the northwest terminated at an area near the Coso geothermal production field[27] with $b > 1.1$ (red). The high-temperature area around this field may have contributed to termination of the rupture and high *b*-values

($b > 1.1$: red). Similar behavior was observed for the 2016 Kumamoto earthquakes[28].

To show that coseismic rupture, which caused stress perturbation along the fault of the $M7.1$ events, played a role in the distribution of post-seismic hazards, the slip distribution of the $M7.1$ quake[23] (Fig. 2f) and the $b$-value distribution based on post-$M7.1$-quake seismicity (Fig. 2e) were compared. An area of low $b$-values ($b < 0.9$: indicative of high stress), colored in blue to purple, within the rectangle shown in Fig. 2e, does not overlap with volumes of high slip ($\geq 3$ m: orange to red) but with volumes that remained unruptured (low slip in Fig. 2f), suggesting that the rupture of this quake released a pronounced amount of overall stress. Note that the rectangle that includes this low-$b$-value area is located not on the Garlock fault but near it. For events falling within this rectangle, the $b$-values show a decrease over time, to values around 0.8. The values are not as low as those immediately before the $M6.4$ and $M7.1$ quakes (Fig. 1b, c, and 2a–c), but contribute the most recent values in a decreasing trend of the $b$-value. Similar to laboratory observations of low and decreasing $b$-values that could previously be detected as a fault of a few centimeters in length that approached failure[12,26], this was found for natural earthquakes with faults tens of kilometers in size.

## Discussion

Given the current tectonic stress that drives the ECSZ[2], it is likely possible to consider that a future activated fault is the one conjugating with the $M7.1$ rupture, as seen by the $M6.4$ and $M7.1$ quake couplet. We calculated changes in Coulomb stress resolved on the $M6.4$-quake-type left-lateral faults at a depth of 10 km (Fig. 3b), where the source faults are the right-lateral rupture of the $M7.1$ quake and the left-lateral rupture of the $M6.4$ quake (for details on the fault models, see Methods). This depth of 10 km was chosen because it is a typical depth of the rectangle with the low-$b$-value zone in Fig. 2e. The changes in stress pull most of the nearby left-lateral-type faults further from failure (blue lobes, namely stress shadows[29]) and push others of the same type closer to it (red lobes). We expect strong stress (red) at the region indicated by the rectangle in the left panel of Fig. 3b. Another possibility of future activation is rupture extension to the southeast: namely, the one along the fault of the $M7.1$ quake. We calculated changes in stress resolved on the $M7.1$-quake-type right-lateral faults, revealing that faults in the zone of low $b$-values are again in an area with stress changes to promote failure (red lobes) in the right panel of Fig. 3b. The same stress-change calculations were conducted for different depths (Supplementary Figs. 11 and 12). The result is not induced by a bias of choice of depth: stress patterns for a depth of 8–12 km covering the rectangle shown in Fig. 2e are similar to each other.

If the zone of currently low $b$-values (Fig. 2e) were more stressed (decrease in $b$-value), seismic activity in this zone would be further enhanced with possibility of future ruptures propagating either along a $M6.4$-quake-type left-lateral fault or along a $M7.1$-quake-type right-lateral fault (Fig. 3b and Supplementary Figs. 11 and 12). If so, the influence of a likely future rupture on the Garlock fault would be inevitable. Although this fault has historically been seismically quiescent, it has hosted numerous large earthquakes over several thousand years[30], and the last major earthquake occurred about 400 to 500 years ago[31]. Moreover, geodetic measurements[1,18,23] showed that measurable surface creep was triggered by the Ridgecrest sequence, while no measurable creep was shown before the start of this sequence[32]. The timing of the precursory signal observed in Fig. 2h remains unexplained: the low-$b$-value patch may continue or subside without the occurrence of a large earthquake. It is not yet possible to make conclusions about the quantitative predictive power of $b$-value mapping. Thus, together with seismological and geodetic observations, it would be worthwhile to monitor the spatio-temporal distribution of $b$-values around the southeast rupture terminus of the $M7.1$ quake, which contributes to seismic hazard in the ECSZ.

A question regarding the finding that the Garlock fault may be at risk of rupture due to the existence of a low $b$-value patch is that the estimate of risk is not quantitative, in the sense of a probability computation. One approach to quantitative evaluation of present level of risk is to apply some type of nowcasting method[33] to the Ridgecrest sequence. While we have not examined it in details, previous studies have shown promise in its applications to seismically active regions[33–36], and on a world-wide basis[37–39]. Our future work will be directed at answering this question.

## Methods

**Earthquake dataset.** The earthquake catalog produced by the SCSN (http://service.scedc.caltech.edu/eq-catalogs/date_mag_loc.php)[9] was used in this study. The SCSN has been in operation for more than 87 years, since 1932, and has recorded and located earthquakes. Station density and technological sophistication have both increased steadily since 1932 leading to increased catalog precision over time.

More than $10^5$ earthquakes with $M \geq 1$ since 1980 at depths shallower than 20 km within the study region shown in Fig. 1a were processed. In view of the network updates, the completeness levels ($M_c \sim 1.5 \pm 0.5$) were obtained in and around the study region since the 1980s[9]. The levels are confirmed in the inset of Fig. 2c and Supplementary Figs. 2a and 7.

For the location of the $M7.1$ quake, the hypocenter obtained from the relocated catalog from SCSN (https://scedc.caltech.edu/research-tools/alt-2011-dd-hauksson-yang-shearer.html)[40,41] was used because the depth of this quake in the relocated catalog (1.9 km) is much shallower than its depth in the standard (non-relocated) catalog (8.0 km). This difference in depth was considered to be non-negligible because the target depth range in cross-sectional views, such as in Fig. 2, is mainly from 0 to 15 km. The relocated catalog was also used for the hypocenter of the $M6.4$ quake, although the difference in depth is small (10.7 km for the routinely generated catalog and 11.8 km for the relocated one).

**$b$-value estimation.** Spatial temporal changes in $b$ are known to reflect a state of stress in the Earth's crust[13,15,42] and to be influenced by asperities and frictional properties[24,43] and by an interface locking along subduction zones[26,44,45]. The results of laboratory experiments indicate a systematic decrease in the $b$-value approaching the time of the entire fracture[11,12,14]. To estimate $b$-values homogeneously over space and time, we employed the EMR (Entire-Magnitude-Range) technique[46], which also simultaneously calculates the $a$-value of the GR law and the completeness magnitude $M_c$, above which all events are considered to be detected by a seismic network (a brief explanation of the EMR technique is provided in the next paragraph). The software package ZMAP[47] was used to facilitate computing and mapping $b$-values based on the EMR method, as described below. EMR applies the maximum-likelihood method when computing the $b$-value to events with a magnitude above $M_c$. A $b$-value was always computed for the corresponding sample only if at least 20 events yielded a good fit to the GR law. Figure 2g shows a good fit of the GR law to observations in the present cases. The top panel of Fig. 2g shows the frequency-magnitude distribution of earthquakes falling within a cylindrical volume with a 5-km radius, centered at the location of the $M6.4$ hypocenter in Fig. 2a, c. The bottom panel shows the same as the top one for the location of the $M7.1$ hypocenter. The significant differences between $b$-values were computed with the Utsu test[48] as the probability $P_b$ that the $b$-values were not different. Values of $\log P_b \leq -1.3$ indicate a significant difference. For both cases in Fig. 2g, the difference in $b$ is significant. This observation is further supported by noting that the absolute difference in $b$ is larger than the sum of uncertainties of the $b$-values for each of the hypocenters, where the uncertainties in $b$, as described in the legend of Fig. 2, were computed by bootstrapping. Uncertainty in $b$ is quantified by the standard deviation of the $b$-values of the bootstrap samples.

The EMR technique[46] was initially proposed by Ogata and Katsura[49,50], who combined the GR law with a detection rate function. Statistical modeling was performed separately for completely detected and incompletely detected parts of the frequency-magnitude distribution. The $b$- and $a$-values in the GR law are computed based on earthquakes above a certain magnitude ($M_{cc}$). For earthquakes whose magnitudes are smaller than $M_{cc}$, it has been hypothesized that the detection rate depends on their magnitudes in such a way that large earthquakes are almost entirely detected while smaller ones are detected at lower rates. Several studies[46,49,50] assumed that the detection rate was expressed by the cumulative function of the Normal distribution. Earthquakes with magnitudes greater than or equal to $M_{cc}$ are assumed to be detected with a detection rate of one. To evaluate

the fitness of the model to data, the log-likelihood is computed by changing the value of $M_{cc}$. The best fitting model is that which maximizes the log-likelihood.

The code of the EMR method is freely available together with the seismicity analysis software package ZMAP (http://www.seismo.ethz.ch/en/research-and-teaching/products-software/software/ZMAP/)[47], which is written in Mathworks' commercial software language Matlab® (http://www.mathworks.com). No knowledge of the Matlab language is needed since ZMAP is GUI-driven, although the ZMAP code is open. ZMAP combines many standard seismological tools. Evaluating spatial variations in seismicity is one of the primary research objectives of ZMAP. By creating a dense spatial grid and sampling overlapping volumes of circular shape, ZMAP users can map $b$-values calculated by the EMR[46]. Throughout this study, a grid spacing of $0.01 \times 0.01°$ for map views (Fig. 1b and Supplementary Fig. 2) and $0.5 \times 0.5$ km for cross-sectional views (Fig. 2 and Supplementary Figs. 4 and 5) was used with a sampling radius $r = 5$ km, except for Supplementary Figs. 1 and 3 created for different radii $r$ to identify the best representatives among them, as described below.

**Mapping procedure.** The optimal sampling volume (Fig. 1b) was searched by mapping $b$-values with a wide range of radii $r$ and the largest radius that provided the most detailed resolution of the $b$-value heterogeneity (inhomogeneity) was selected. The observation of a nearly identical pattern of $b$-values when sampled with radii of $r = 5$ km and 7 km suggests that using $r < 5$ km only reduces coverage (Supplementary Fig. 1). Sampling with $r \geq 9$ km results in smoothed $b$-values and obscures any $b$-value contrast. Thus, the appropriate radius of the volumes is about 5 or 7 km, because sampling with smaller radii reduces coverage while sampling with larger radii obscures anomalies and contrasts. In making $b$-value maps throughout this study, earthquakes within a radius of $r = 5$ km (Fig. 1b and Supplementary Fig. 2c), a small radius between the appropriate radii, were sampled. The EMR technique[46] also calculates $M_c$ and $a$ simultaneously, thus the maps of $M_c$ and $a$, which were created when the $b$-value map in Fig. 1b and Supplementary Fig. 2c was obtained, are shown in Supplementary Fig. 2a, b. A similar search was conducted for cross-sectional views (Supplementary Fig. 3) and a decision was made to sample earthquakes within a radius of $r = 5$ km (Fig. 2), the same radius used for the map view in Fig. 1b and Supplementary Fig. 2c.

A $b$-value analysis is critically dependent on a robust estimate of completeness of the processed earthquake data. In particular, underestimates in $M_c$ lead to systematic underestimates in $b$-values. Attention was always paid to $M_c$ when assessing $M_c$ locally at each node. On the other hand, it is of interest to understand how $M_c$ factors into the conclusion, so an additional test was conducted. This was achieved by creating $b$-value cross-sections and timeseries for an increased value of every local $M_c$ by 0.2 and 0.5 magnitude units[44] (Supplementary Figs. 4 and 5). The spatial and temporal pattern in $b$ generally appears to remain stable with the $M_c$ correction. However, due to a reduced plotted area, it was not possible to judge whether the predictive information in the $b$-value is contained in the very smallest earthquakes when using small values for the $M_c$ correction or in the intermediate magnitude events when using large values. Future research will be to tackle this problem, using a seismicity catalog similar to that including highly abundant earthquakes, derived from template matching[1].

**Fault models.** The fault model of the $M7.1$ quake, which was used in this study, is based on the work by Xu et al.[23], one of the currently available fault models of this quake. This is a finite-fault model (also called variable slip) with numerous small patches of slip, each having information on the location of a rectangular patch, strike, dip, and rake. Details of this fault model were obtained via the website created by the same authors (https://topex.ucsd.edu/SV_7.1/index.html)[23] and by accessing the Earthquake Source Model Database (SRCMOD), an online database of finite-fault rupture models of past earthquakes (http://equake-rc.info/srcmod/)[51]. The model represents a northwesterly striking fault with right-lateral planes, showing one main rupture with two sub-parallel strands near the $M7.1$ hypocenter, to match the Earth's surface deformation imaged by the Interferometric Synthetic Aperture Radar (InSAR) and others. Note that data obtained by using these imaging tools show exquisite details in the near field of earthquake rupture, in contrast to using tele-seismic imaging ones. Considering the location of the hypocenter, the rupture radiation was bilateral. A slip distribution for the main rupture with peak-slip values of 4-5 m is given in Fig. 2f. Average strike, dip, and rake over patches for the main-rupture fault are 136°, 90°, and 176°, respectively, where the standard Aki & Richards sign conventions for fault geometry and slip are used[52]. The model involves predominantly strike-slip faulting.

Similarly, the finite-fault model of the $M6.4$ quake, again proposed by Xu et al.[23], was used. This fault was southwesterly striking with a left-lateral plane. The strike and dip of the plane are 226° and 90°, respectively, and average rake over patches is 6°, revealing a predominant strike-slip fault. A slip distribution with peak-slip values of ~3 m is given in https://topex.ucsd.edu/SV_7.1/index.html[23] and http://equake-rc.info/srcmod/[51]. A previous observation[1] shows that the $M6.4$ quake ruptured conjugate faults that were northwest- and southwest-trending, but the southwest-trending fault was assumed for the $M6.4$ quake, as described in the main text. Then, the model[23] was assumed to be applicable for this fault.

A moment tensor solution based on seismograms recorded by the SCSN (http://service.scedc.caltech.edu/eq-catalogs/date_mag_loc.php)[9] was used to define $M \geq 4.5$ events during the period between the $M6.4$ and $M7.1$ quakes as source faults

of Coulomb stress changes resolved on the fault of the $M7.1$ quake (Fig. 3a and Supplementary Fig. 6). To ensure quality of the mechanisms, solutions for events with a reduction in variance >80%, and generated by using at least three stations, were used. Solutions for two $M5$-class ($M5.0$ and $M5.4$) and five $M4.5$-class events met this criterion. The moment tensor catalog contains two nodal planes. A plane whose strike better matched the lineation in seismicity was chosen.

**Stress-change calculation.** Static stress changes caused by the displacement of a fault (source fault) were calculated (Fig. 3 and Supplementary Figs. 6, 11, 12). Displacements in the elastic half space were used to calculate the 3D strain field; this was multiplied by elastic stiffness to derive the stress changes. A typical value for Poisson's ratio ($PR = 0.25$), Young's modulus ($E = 8 \times 10^5$ bar), and friction coefficient ($\mu = 0.4$) was used, resulting in a shear modulus of $G = E/[2(1+PR)] = 3.3 \times 10^5$ bar. The shear and normal components of the stress change were resolved on specified receiver fault planes. A receiver fault consists of planes, each characterized by specified strike, dip, and rake, on which the stresses imparted by the source faults were resolved. The Coulomb failure criterion, in which failure is hypothesized to be promoted (inhibited) when the Coulomb failure is positive (negative), was used. Coulomb[16,17], the graphic-rich deformation and stress-change software for earthquakes, tectonics, and volcanoes, was used to calculate how earthquakes promote or inhibit failure on nearby faults (https://earthquake.usgs.gov/research/software/coulomb/).

For the fault ruptured by the $M7.1$ quake (Fig. 3b), the finite-fault model proposed by Xu et al.[23] was used (for details, see the section titled Fault models). The fault model of the $M6.4$ quake, again proposed by the same authors[23], was used. These fault models were combined to create source faults in the Coulomb stress-change calculation in Fig. 3b and Supplementary Figs. 11 and 12. When slip on all patches consisting of a finite-fault model is set to zero, the model with zero slip can be used for a receiver fault. In creating Fig. 3a and Supplementary Fig. 6, which show stress changes resolved on the $M7.1$-quake fault, slip on all patches consisting of this fault was set to zero.

To calculate Coulomb stress changes, shown in Fig. 3a and Supplementary Fig. 6, imparted by the $M6.4$ quake and the following $M \geq 4.5$ events up to the $M7.1$ quake, the $M \geq 4.5$ events needed to be modeled. The SCSN moment tensor data, as described in the section titled Fault models, were used. To make realistically scaled source faults from the moment tensor information, a subroutine program in Coulomb was used[16,17]. The faults of the $M \geq 4.5$ events built by using this program were projected onto the Earth's surface, and these are shown in the inset of Fig. 3a and Supplementary Fig. 6i, where the $M6.4$-quake fault is also included. An alternative perspective view to show the faults of the $M \geq 4.5$ events and the $M6.4$-quake is given in Supplementary Fig. 6.

**Homogeneity of seismicity recordings.** Small events in clusters such as swarms, aftershocks, and foreshocks are often missed in the earthquake catalog, as they are masked by the coda of large events and overlap with each other on seismograms. According to previous cases[46,53], $M_c$ depends on time $t$. In creating the inset of Fig. 2c and Supplementary Fig. 7a, a moving window approach was used, whereby the window covered 100 events. $M_c$-values from temporal analysis were plotted at the end of the moving window that they represent. $M_c$ decreased with $t$ from nearly 3 and reached a constant value at around 1.6. Relatively large events occurred early in the sequence, and the mean magnitude of these events evolved to small values over time (grey). The time-dependent decrease in $M_c$ is consistent with the data. The use of $M \geq 3$ events secures the homogeneity of recording for temporal analysis of seismicity after the $M6.4$ quake.

We compared the EMR method with two frequently-used techniques (Supplementary Fig. 7): Maximum-curvature (MAXC) method and Goodness-of-fit (GOF) method[54]. The MAXC method defines $M_c$ as the magnitude bin with the highest number of events (red). GOF defines $M_c$ as the lowest magnitude for which the GOF is 90% or larger (green), where the GOF is based on the residual indicating the deviance of the fitted GR law model from the observed data[54]. Both techniques underestimated $M_c$ (Supplementary Fig. 7a), that is, the EMR method gave the most conservative estimation of $M_c$, which justifies the use of this method. The same was conducted as Supplementary Fig. 7a for seismicity from 1 January 1980 to immediately before the $M6.4$ quake along the fault ruptured by the $M7.3$ quake (from $P_1$ to $P_2$), obtaining the same feature (Supplementary Fig. 7b).

**OU law.** An exact expression of the OU law[19] is given as $\lambda = k(c + t)^{-p}$, where $t$, $\lambda$, $p$ are the same as those defined in the main text, and $c$ and $k$ are constants. Similar to the GR case, the maximum-likelihood fit was used to determine the parameters for this law. Uncertainties in $p$ were computed by bootstrapping, where the standard deviation of the $p$-values of the bootstrap samples quantifies the corresponding uncertainty in $p$. The codes of the OU-fitting method are available together with the seismicity analysis software package ZMAP[47]. Similar to the observation that $p = 1$ is generally a good approximation[20], a theory employing the rate- and state-dependent friction law[21] also assumes $p = 1$ as a standard form. Variability in $p$ is possible: $p > 1$ and $p < 1$ for special cases with rapidly and slowly decreasing stress, respectively.

The inset of Fig. 4 shows a good fit of the OU law to activity with $M \geq 3$ for the entire area along the $M7.1$ rupture (from $P_1$ to $P_2$) during the period between the

$M6.4$ and $M7.1$ quakes ($t = 0–1.404$ days: 34 h). Note that setting of a minimum magnitude at $M = 3$ ensures the homogeneity of recording after the $M6.4$ quake (for details, see the section titled Homogeneity of seismicity recordings, inset of Fig. 2c, and Supplementary Fig. 7a).

Seismicity along the $M7.1$ rupture was divided into two areas (Fig. 4): northern area (north of 35.72°N, blue data) and southern area (south of 35.72°N, red data), where the northern area includes the $M7.1$ hypocenter. Several lengths of the analyzed period were also considered. The difference in $p$ between these areas was insignificant for periods shorter than 1 day, beyond which $p$-values for the southern area were significantly larger than for the northern one. $p$-values for the northern area were smaller than the typical decay with $p \sim 1$[20] if unreliable estimates (open circles) were not taken into consideration. $p$-values for the southern area showed $p > 1$ if the longest period (1.404 days) was considered. The result in Fig. 4 was not induced by model selection bias: using the ETAS (Epidemic-type Aftershock Sequence) model[55,56], similar results were observed (Supplementary Fig. 8), as described below. A model for seismicity rate resulting from stressing history[21] explains the result in Fig. 4 and Supplementary Fig. 8: although the overall trend in stress decreased with time, low $p$-values ($p < 1$) in the northern area were due to slower decreasing stress than in the southern area.

A more sophisticated model such as the ETAS model[55] can be used, instead of the OU model. It was examined whether the ETAS model provided similar results to those obtained by the OU model, using the package SASeis2006[56] for facilitating statistical analysis of seismicity (https://www.ism.ac.jp/editsec/csm/index_j.html). This package includes a program that can be used to fit the ETAS model to earthquake (aftershock) data. The package also provides modules for plotting figures. Supplementary Figs. 8c and 8d showing a good fit of the ETAS model to observations in the present cases, were created by using one of the modules. For fitting the ETAS model, two time intervals were considered. One interval is called the target interval for which the ETAS model parameters are computed. The seismicity in this period may be affected by earthquakes which occurred before this period due to the long-lived nature of aftershock activity. To consider this effect, the other time interval that is precursory to the target interval (called precursory interval) was chosen and aftershock activities following earthquakes in this period were considered in the computation. The interval between the dashed lines in Supplementary Figs. 8c and 8d is the target interval (0.001 to 1.404 days) for which the ETAS model parameters were computed, and the precursory interval is 0-0.001 days. The correlation between $p$ and the length of the analyzed period was independent of the choice of model for estimating $p$-values (Fig. 4 and Supplementary Fig. 8a). This suggests that the OU-based approach (Fig. 4) is sufficient to capture the essential aspects of the relaxation process that followed the $M6.4$ earthquake. Supplementary Fig. 8b shows a case that uses slightly different target intervals and precursory intervals from those in Supplementary Fig. 8a, obtaining a similar result.

**Spatial organization.** A previous study[22] using the SCSN earthquake catalog showed that an observation on seismic concentration and dispersion defined by $\phi$-values was used to discriminate spatial clustering due to retrospectively named foreshocks from the one induced by aftershocks, and was implemented in an alarm-based model to forecast $M > 6$ earthquakes. Moreover, the probability of a daily occurrence presented an isolated peak due to a concentration of seismicity closely located in time and space to the epicenter of 5 out of 6 $M > 6$ earthquakes. A comparison with the present study shows that seismicity after the $M6.4$ quake displayed a foreshock-type sequence ($\phi > 1$) in a region near the eventual $M7.1$ hypocenter and an aftershock-type sequence in other regions.

The previous study[22] that introduced $\phi = R^{-1}/R_b^{-1}$ to seismicity analysis did not take its uncertainty into consideration. A new uncertainty assessment based on a bootstrap method was thus developed. This approach is a Monte Carlo style simulation based on catalogs with permuted distances from position $x$ to events. For each of the locations $x$ where $\phi$ needs to be calculated, two catalogs are required: one consisting of $n$ events to be used for calculating $R^{-1}$ and the other consisting of $n$ events for $R_b^{-1}$ calculation. For the former catalog, events are drawn $n$ times from its population of $n$ events, allowing any event to be selected more than once. From these events, $R^{-1}$ is computed, as defined in the main text. The same applies to $R_b^{-1}$ to finally compute $\phi$. This process was repeated 300 times, and their errors were estimated as the standard deviation of the $\phi$-values, $\sigma_\phi$.

$\sigma_\phi$ with different values for $n$ onto two cross-sections (from $P_1$ to $P_2$, and from $P_3$ to $P_4$) for the time immediately before the $M7.1$ quake was mapped, as shown in Supplementary Fig. 10. $\sigma_\phi$-values were also used to draw an error bar at each data point in the $\phi$-value timeseries (Fig. 5c, d and Supplementary Fig. 10). General features are that high $\sigma_\phi$-values ($\sigma_\phi > 0.2$) fall in regions with high $\phi$-values and that the $\sigma_\phi$-value increases with a decrease of $n$-value. Using the newly developed uncertainty assessment, the significance of the results was quantified. Taking its uncertainty ($\sigma_\phi$) into consideration, $\phi$-values in a region near the $M7.1$ hypocenter can be larger than 1 (seismic concentration) at the time immediately before the $M7.1$ quake, while $\phi$-values in other regions are $\phi \sim 1$ or $\phi < 1$. Cases for $n = 15$-35 show that the high $\phi$-value anomaly near the $M7.1$ hypocenter is stable and significant. A case with $n = 10$ shows an unstable and insignificant result due to small populations, resulting in high $\sigma_\phi$-values.

## Data availability
The SCSN earthquake catalog[9] used in this study is available at http://service.scedc.caltech.edu/eq-catalogs/date_mag_loc.php. In creating map images in Figs. 1 and 3 and Supplementary Figs. 1, 2, 11, and 12, active fault data provided in the software Coulomb were used[16,17]. Fault models of the $M7.1$ and $M6.4$ quakes were obtained from https://topex.ucsd.edu/SV_7.1/index.html[23] and SRCMOD (http://equake-rc.info/srcmod/)[51]. Hypocenters of these quakes were obtained from the Southern California Earthquake Data Center, Special Data Sets (https://scedc.caltech.edu/research-tools/alt-2011-dd-hauksson-yang-shearer.html)[40,41]. The data that support the finding of this study are available from the corresponding author upon reasonable request.

## Code availability
The seismicity analysis software package ZMAP[47], used for Figs. 1 and 2 and Supplementary Figs. 1–5, and 7, was obtained from http://www.seismo.ethz.ch/en/research-and-teaching/products-software/software/ZMAP. The graphic-rich deformation and stress-change software Coulomb[16,17], used for Fig. 3 and Supplementary Figs. 6, 11, and 12, is available for download at https://earthquake.usgs.gov/research/software/coulomb/. The program SASeis2006[56] (Statistical Analysis of Seismicity-updated version), used for Fig. 4 and Supplementary Fig. 8, was obtained from https://www.ism.ac.jp/editsec/csm/index_j.html. The Generic Mapping Tools (GMT)[57], used for Fig. 1 and Supplementary Figs. 1 and 2, are an open-source collection (https://www.generic-mapping-tools.org). The programs used for Fig. 5 and Supplementary Figs. 9 and 10, are available from the corresponding author upon reasonable request.

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

## Acknowledgements

The author thanks M. Kamogawa, R. Shcherbakov, and S. Toda for discussion. Some figures were produced by using GMT software[57]. This study was partially supported by the Ministry of Education, Culture, Sports, Science and Technology (MEXT) of Japan, under its The Second Earthquake and Volcano Hazards Observation and Research Program (Earthquake and Volcano Hazard Reduction Research) and by JSPS KAKENHI Grant Number JP 20K05050.

## Author contributions

K.Z.N. conceived the analysis method, conducted data analysis, created figures, and wrote the paper.

## Competing interests

The author declares no competing interests.
