## [Peer Review File · Nature Communications]

Reviewers' comments:

Reviewer #1 (Remarks to the Author):

This is a very readable paper, linking the local and time dependent variation in seismic b-value before and after the Ridgecrest earthquakes with estimates of the stress changes as computed from the program Coulomb. The results indicate that there is predictive information in the b-value obtained which are the only actual observables that one has.

A few observations. I would be curious to understand how the magnitude cutoff plays into the conclusions. That is, whether the information is contained in the very smallest earthquakes, or in the intermediate magnitude events. I would also curious to know how the events of Ridgecrest are related to events earlier in time, including the 1999 Hector Mine sequence and the 1992 Landers event. But possibly this is a calculation for another paper. In any event, I think this paper can be published substantially as is. The English is good with the exception of a few minor and obvious corrections needed.

Reviewer #2 (Remarks to the Author):

Summary:

This study by K. Nanjo investigates earthquake triggering and characteristics of seismicity before, during, and after the 2019 Ridgecrest earthquakes. Nanjo focuses in particular on determining maps of "b-values" for different time periods, and shows how the nucleation area for the Mw 6.4 foreshock and Mw 7.1 mainshock both had low b-values before these events occurred, and mid-to-high b-values afterward. Nanjo shows that the b-value map correlates well with the slip distribution of the mainshock. I think the main conclusions are that the b-values provide insights into the state of stress in the fault zone, which is likely closely related to the nucleation and evolution of earthquakes in the sequence. I think this is an interesting paper and has a generally compelling analysis. I think the topic is appropriate for the journal.

Questions/comments:

- 1) The author refers to the supplementary material frequently, and since there are only 3 main text figures, I feel like some of these should be moved from the supplement to the main text. My understanding is that the format of this journal can accommodate more figures.
- 2) Lines 134-139 discuss a supplementary figure, which relies on a metric that is not commonly used in earthquake science (if ever). The figure and text do not make sense without digging into the methods to figure out what you are talking about.
- 3) The quantity ϕ defined by the author in the Spatial Organization section should be reported with uncertainties. More discussion is needed about this, why it was chosen.
- 4) The points about the Garlock fault and aseismic behavior are probably not relevant here. For one, none of the seismicity you are looking at is on the Garlock fault where it was creeping. Also, this creep is only at very shallow depths, compared to any seismicity in the area, which is much deeper. Finally the argument about low b-values reflecting aseismic potential of the fault I think is incorrect, as this ignores the frictional stability ($a-b$) of the fault itself. You could have a rate strengthening fault with high stress and the fault would still creep.
- 5) The slip model for the mainshock is purely teleseismic and therefore does not include any of the high quality near-field data. This is important since you are calculating Coulomb stresses in the near-field. Also, rather than using a slip model for the Mw 6.4, the author instead seems to take a

moment tensor from the SCSN catalog, and somehow apply this to a discretized grid. This makes a lot of assumptions and is problematic. The analysis should be repeated using one of the slip models already published for both events.

6) I'm not sure why the Coulomb analysis was only done for the NW part of the rupture. There is definitely slip on the SE part.

7) The mainshock location used by Nanjo is not the best hypocenter provided by the SCSN. This location is just using a 1D velocity model. I would recommend using the location provided here by Hauksson: <https://scedc.caltech.edu/research-tools/alt-2011-dd-hauksson-yang-shearer.html> . It looks generally like the mainshock depth is very shallow, around 2 km. This seems to make your story more consistent, actually.

8) Los Angeles is spelled incorrectly in the figure 1 caption

9) Subfigure letters e.g. (A) should be in bold on figures to ease visibility and match the captions.

Response to reviewers' comments

Below please find the comments from the two reviewers (black) as well as my response to them (blue). At the beginning of each comment from Reviewer #1, I added a number, such as R1-1 and R1-2. Similarly, I added R2-1, R2-2, etc. to reflect responses to comments from Reviewer #2. The abbreviations used below are OM: original manuscript and RM: revised manuscript. Major changes that were made have been highlighted in red in the main text. I wish to thank both reviewers for these very valuable comments and suggestions, and trust that the edits made have fully addressed their concerns and requests.

Reviewer #1 (Remarks to the Author):

R1-1. This is a very readable paper, linking the local and time dependent variation in seismic b-value before and after the Ridgecrest earthquakes with estimates of the stress changes as computed from the program Coulomb. The results indicate that there is predictive information in the b-value obtained which are the only actual observables that one has.

- Thank you for this comment, which was discussed in the main text (Page 3).

R1-2. A few observations. I would be curious to understand how the magnitude cutoff plays into the conclusions. That is, whether the information is contained in the very

smallest earthquakes, or in the intermediate magnitude events. I would also be curious to know how the events of Ridgecrest are related to events earlier in time, including the 1999 Hector Mine sequence and the 1992 Landers event. But possibly this is a calculation for another paper. In any event, I think this paper can be published substantially as is. The English is good with the exception of a few minor and obvious corrections needed.

- According to these comments, I agreed that it is of interest to understand how M_c plays into the conclusion (Page 12). So, I conducted an additional test. This was achieved by creating b -value cross-sections and timeseries for an increased value of every local M_c by 0.2 and 0.5 magnitude units, as was done by Nanjo & Yoshida [2018] (Supplementary Figs. 4 and 5). It seems that the spatial and temporal pattern in b generally remains stable, even with an M_c correction. However, due to a reduced plotted area, it was not possible to judge whether the predictive information in the b -value is contained in the very smallest earthquakes when using small values for M_c correction or in the intermediate magnitude events when using large ones. One possibility is that the data used here are not sufficiently abundant for the preset test. Future research will be to tackle this problem, using a seismicity catalog similar to that including highly abundant earthquakes, derived from template matching [Ross et al., 2019].
- In response to another part of the reviewer's comment, several papers [Pope & Mooney, 2019; Stein & Sevilgen 2019; Stein et al., 2019] were referenced and a description how the events of Ridgecrest are related to events earlier in time was provided, as follows (Page 2): *Coulomb stress models were used to explain that the site of the M6.4 quake was stressed by the great 1872 Owens Valley (M~7.6), the*

1992 Landers (M7.3), and the 1999 Hector Mine (M7.1) quakes, and that the M6.4 earthquake loaded the site where the M7.1 shock nucleated.

- According to the last comment, the current manuscript was carefully proofread to correct all English language errors.

Reviewer #2 (Remarks to the Author):

R2-1. This study by K. Nanjo investigates earthquake triggering and characteristics of seismicity before, during, and after the 2019 Ridgecrest earthquakes. Nanjo focuses in particular on determining maps of “b-values” for different time periods, and shows how the nucleation area for the Mw 6.4 foreshock and Mw 7.1 mainshock both had low b-values before these events occurred, and mid-to-high b-values afterward. Nanjo shows that the b-value map correlates well with the slip distribution of the mainshock. I think the main conclusions are that the b-values provide insights into the state of stress in the fault zone, which is likely closely related to the nucleation and evolution of earthquakes in the sequence. I think this is an interesting paper and has a generally compelling analysis. I think the topic is appropriate for the journal.

- Based on this comment, additional discussion was added to the main text (Page 3).

Questions/comments:

R2-2. 1) The author refers to the supplementary material frequently, and since there are only 3 main text figures, I feel like some of these should be moved from the supplement to the main text. My understanding is that the format of this journal can accommodate

more figures.

- I thank the reviewer for this valuable comment. Figures of fitting of the OU law and seismicity concentration/dispersion that were shown in the Supplementary Information of the OM are now shown as Figs. 4 and 5 (Pages 30 and 31) in the main text of the RM. These figures are described in the main text (Pages 5 and 6). Thus, supplementary material is less frequently referred to.

R2-3. 2) Lines 134-139 discuss a supplementary figure, which relies on a metric that is not commonly used in earthquake science (if ever). The figure and text do not make sense without digging into the methods to figure out what you are talking about.

- Based on this useful comment, the main text was modified (Page 6) and showed Fig. 5, which is associated with this metric (Page 31). I hope that this modification satisfies Reviewer #2.

R2-4. 3) The quantity ϕ defined by the author in the Spatial Organization section should be reported with uncertainties. More discussion is needed about this, why it was chosen.

- This is an excellent suggestion. I defined uncertainties of ϕ (σ_ϕ) and reported them in the main text (Page 17), in Fig. 5 (Page 31), and in Supplementary Fig. 10. Taking σ_ϕ into consideration, I showed that ϕ -values in a region near the *M7.1* hypocenter can be larger than 1 at the time immediately before the *M7.1* quake, while ϕ -values in other regions are $\phi \sim 1$ or $\phi < 1$.
- In response to another part of the reviewer's comment, I discussed why ϕ was chosen (Page 6). I first described that to support the observation that the events

preceding the $M7.1$ quake very likely played a role in triggering the eventual $M7.1$ event, an independent analysis from the stress-related analyses in Figs. 1-4 (Pages 25-30) was conducted. Next, I pointed out that this was achieved by investigating if any sign indicative of the $M7.1$ quake could be found in the spatial organization in seismicity after the $M6.4$ quake. Referring to Lippiello et al. [2012], I discussed that the spatial concentration of smaller magnitude events (retrospectively named foreshocks) near the eventual event (retrospectively named mainshock) was a common feature with large earthquakes in southern California. To examine whether this was observed for the $M7.1$ quake, I chose ϕ , as was done by Lippiello et al. [2012]. I hope that this discussion satisfies Reviewer #2.

R2-5. 4) The points about the Garlock fault and aseismic behavior are probably not relevant here. For one, none of the seismicity you are looking at is on the Garlock fault where it was creeping. Also, this creep is only at very shallow depths, compared to any seismicity in the area, which is much deeper. Finally the argument about low b-values reflecting aseismic potential of the fault I think is incorrect, as this ignores the frictional stability (a-b) of the fault itself. You could have a rate strengthening fault with high stress and the fault would still creep.

- I understood the reviewer's comment, and removed the related discussion.

R2-6. 5) The slip model for the mainshock is purely teleseismic and therefore does not include any of the high quality near-field data. This is important since you are calculating Coulomb stresses in the near-field. Also, rather than using a slip model for the M_w 6.4, the author instead seems to take a moment tensor from the SCSN catalog,

and somehow apply this to a discretized grid. This makes a lot of assumptions and is problematic. The analysis should be repeated using one of the slip models already published for both events.

- Based on this suggestion, slip models for the *M*6.4 and *M*7.1 quakes were used, as proposed by Xu et al. [2020], as they are some of the currently available slip models that include high quality near-field data (Page 12). See Fig. 3 (Page 28), Supplementary Figs. 6, 11, and 12.

R2-7. 6) I'm not sure why the Coulomb analysis was only done for the NW part of the rupture. There is definitely slip on the SE part.

- I appreciate the reviewer's comment. The Coulomb analysis was performed not only for the NW part of the rupture but also for the SE part of it in Fig. 3a (Page 28).

R2-8. 7) The mainshock location used by Nanjo is not the best hypocenter provided by the SCSN. This location is just using a 1D velocity model. I would recommend using the location provided here by Hauksson: <https://scedc.caltech.edu/research-tools/alt-2011-dd-hauksson-yang-shearer.html>. It looks generally like the mainshock depth is very shallow, around 2 km. This seems to make your story more consistent, actually.

- In fact, I used the hypocenter obtained from the relocated catalog for the location of the *M*7.1 quake (Page 9). I agree that this fortifies the consistency of the story.

R2-9. 8) Los Angeles is spelled incorrectly in the figure 1 caption

- Thank you for that comment. Los Angeles is now spelled correctly (Fig. 1, Page 25).

R2-10. 9) Subfigure letters e.g. (A) should be in bold on figures to ease visibility and match the captions.

- According to the comment, subfigure letters are now in bold in figures. Please see all figures.

References

- Lippiello, E., Marzocchi, W., de Arcangelis, L. & Godano, C. Spatial organization of foreshocks as a tool to forecast large earthquakes. *Scientific Reports* 2, 846 (2012).
- Nanjo, K. Z. & Yoshida, A. A b map implying the first eastern rupture of the Nankai Trough earthquakes. *Nature Communications* **9**(1), 1117 (2018).
- Pope, N. H. & Mooney, W. Coulomb stress models for the 2019 Ridgecrest earthquake sequence, California. American Geophysical Union, Fall Meeting 2019, abstract #S31G-0507 (2019).
- Ross, Z. E., Idini, B., Jia, Z., Stephenson, O. L., Zhong, M., Wang, X., Zhan, Z., Simons, M., Fielding, E. J., Yun, S.-H., Hauksson, E., Moore, A. W., Liu, Z. & Jung, J. Hierarchical interlocked orthogonal faulting in the 2019 Ridgecrest earthquake sequence. *Science* 366(6463), 346-351 (2019).
- Stein, R. S. & Sevilgen, V. Southern California M 6.4 earthquake stressed by two large historic ruptures. *Temblor*, <http://doi.org/10.32858/temblor.034> (2019).
- Stein, R. S., Hobbs, T., Rollins, C., Ely, G., Sevilgen, V. & Toda, S. Magnitude 7.1

earthquake rips northwest from the M6.4 just 34 hours later. *Temblor*, <http://doi.org/10.32858/temblor.037> (2019).

- Xu, X., Sandwell, D. T. & Smith-Konter, B. Coseismic displacements and surface fractures from Sentinel-1 InSAR: 2019 Ridgecrest earthquakes. *Seismological Research Letters* doi: 10.1785/0220190275 (2020).

REVIEWERS' COMMENTS:

Reviewer #1 (Remarks to the Author):

According to the authors, "Crustal deformation due to the July 2019 earthquake sequence in Ridgecrest (California) that culminated in a preceding quake of magnitude (M) 6.4 and a subsequent M7.1 quake caused stress perturbation in the nearby region, but it is yet uncertain about implication of future seismic activity. We investigated the occurrence of small earthquakes compared to larger ones, the b-values, and found that the rupture initiation from an area of low-b-values, indicative of high stress, was common to both M6.4 and M7.1 quakes. The post-M7.1-quake sequence revealed that another low-b-value zone, which avoided its ruptured area, fell into the remaining unruptured area. This shows that if a high-likelihood future rupture were to occur, this might influence the nearby Garlock fault that hosted large earthquakes for several thousand years."

This is a very good paper that outlines the science of b-value mapping, and provides a nice summary of the method. Applying it to the region of the Ridgecrest earthquake, they find that the Garlock fault may be at risk of rupture due to the existence of a low b-value region. And while the estimate of risk is not quantitative, in the sense of a probability computation, the results are nonetheless interesting and worthy of publication.

I would urge the authors to consider how one might compute a probability of future large earthquake in the region, or perhaps apply some type of nowcasting method to evaluate present level of risk. Aside from that comment, I think the paper can be published substantially as is.

Reviewer #2 (Remarks to the Author):

The author has addressed my concerns appropriately.

Response to reviewer's comments

Below please find the comments from the two reviewers (black) as well as my response to them (blue). I wish to thank the Reviewer #1 for her/his very valuable comments and suggestions, and trust that the edits made have fully addressed her/his concerns and requests.

Reviewer #1 (Remarks to the Author):

This is a very good paper that outlines the science of b-value mapping, and provides a nice summary of the method. Applying it to the region of the Ridgecrest earthquake, they find that the Garlock fault may be at risk of rupture due to the existence of a low b-value region. And while the estimate of risk is not quantitative, in the sense of a probability computation, the results are nonetheless interesting and worthy of publication.

I would urge the authors to consider how one might compute a probability of future large earthquake in the region, or perhaps apply some type of nowcasting method to evaluate present level of risk. Aside from that comment, I think the paper can be published substantially as is.

- Thank you for this comment, which was discussed in the main text as follows (Lines of 253-259 of Page 9): *A question regarding the finding that the Garlock fault may be at risk of rupture due to the existence of a low b-value patch is that the*

estimate of risk is not quantitative, in the sense of a probability computation. One approach to quantitative evaluation of present level of risk is to apply some type of nowcasting method³³ to the Ridgecrest sequence. While we have not examined it in details, previous studies have shown promise in its applications to seismically active regions³³⁻³⁶, and on a worldwide basis³⁷⁻³⁹. Our future work will be directed at answering this question. I hope that this discussion satisfies Reviewer #1.

References

33. Rundle, J. B., Turcotte, D. L., Donnellan, A., Grant Ludwig, L., Luginbuhl, M. & Gong, G. Nowcasting earthquakes. *Earth and Space Science* 3(11), 480-486 (2016).
34. Luginbuhl, M., Rundle, J. B. & Turcotte, D. L. Statistical physics models for aftershocks and induced seismicity. *Philosophical Transactions of the Royal Society A* 377, 20170397 (2018).
35. Fildes, R. A., Kellogg, L. H., Turcotte, D. L. & Rundle, J. B. Interevent seismicity statistics associated with the 2018 quasiperiodic collapse events at Kīlauea, HI, USA. *Earth and Space Science* 7(3), e2019EA000766 (2020).
36. Rundle, J. B. & Donnellan, A. Nowcasting earthquakes in southern California with machine learning: Bursts, swarms and aftershocks may reveal the regional tectonic stress. *Earth and Space Science*, <https://doi.org/10.1002/essoar.10501945.1> (2020).
37. Rundle, J. B., Luginbuhl, M., Giguere, A. & Turcotte, D. L. Natural time, nowcasting and the physics of earthquakes: Estimation of seismic risk to global megacities. *Pure and Applied Geophysics* 175, 647-660 (2018).
38. Rundle, J. B., Giguere, A., Turcotte, D. L., Crutchfield, J. P. & Donnellan, A. Global seismic nowcasting with Shannon information entropy. *Earth and Space Science*

6(1), 191-197 (2019).

39. Rundle, J. B., Luginbuhl, M., Khapikova, P., Turcotte, D. L., Donnellan, A. & McKim, G. Nowcasting Great Global Earthquake and Tsunami Sources. *Pure and Applied Geophysics* 177, 359-368 (2020).